# Aortic Root Replacement Procedures: A Validation Study of the Western Denmark Heart Registry from 1999 to 2022

**DOI:** 10.3390/diagnostics15050611

**Published:** 2025-03-04

**Authors:** Emil Johannes Ravn, Viktor Poulsen, Poul Erik Mortensen, Jordi Sanchez Dahl, Kristian Øvrehus, Oke Gerke, Ivy Susanne Modrau, Katrine Müllertz, Lars Peter Schødt Riber, Lytfi Krasniqi

**Affiliations:** 1Department of Cardiothoracic Surgery, Odense University Hospital, 5000 Odense, Denmark; emil.johannes.ravn@rsyd.dk (E.J.R.);; 2Department of Cardiology, Odense University Hospital, 5000 Odense, Denmark; 3Department of Clinical Research, University of Southern Denmark, 5230 Odense, Denmark; oke.gerke@rsyd.dk; 4Department of Nuclear Medicine, Odense University Hospital, 5000 Odense, Denmark; 5Department of Cardiac, Thoracic and Vascular Surgery, Aarhus University Hospital, 8200 Aarhus, Denmark; 6Department of Clinical Medicine, Aarhus University, 8000 Aarhus, Denmark; 7Department of Cardiology, Aalborg University Hospital, 9000 Aalborg, Denmark

**Keywords:** aortic root replacement, Western Danish Heart Registry, epidemiology

## Abstract

**Background/Objectives**: We reviewed data from the Western Danish Heart Registry (WDHR), which collects mandatory information on heart surgeries in Western Denmark, to validate cases with aortic root replacement (ARR) and assess the validity of registered data for all recorded cases. **Methods**: Patients registered in the WDHR with Danish Health Care Classification System (SKS) codes KFC and KFM from January 1999 to April 2022 were reviewed using electronic medical records. All patients who underwent ARR were included, and clinical data from the WDHR were adjudicated against electronic medical records. **Results**: A total of 847 cases with ARR were identified. Missing values averaged 12.0% in baseline variables (range: 3.2–22.1%), 7.3% in EuroSCORE II variables (range: 0.8–48.9%), and 5.5% in postoperative outcome variables (range: 4.1–8.1%). After adjudication, unrecovered data averaged 6.5% for baseline variables (range: 0.1–11.7%), 5.3% for EuroSCORE II variables (range: 0–32.5%), and 0.5% for postoperative outcomes (range: 0–0.8%). Missing data among EuroSCORE II were lower from 2012 and beyond (2.9% (range: 0.6–14.3%)). The median EuroSCORE II according to the WDHR was 6.2% (95% confidence interval 1.4–6.3) and after adjudication 10.7% (95% confidence interval 3.3–13.3). The positive predictive value for arrhythmia, central nervous damage, dialysis, reoperation for bleeding, and reoperation for ischemia exceeded 95%. **Conclusions**: The WDHR demonstrated overall value for clinical epidemiological research in ARR, but cases require validation to differentiate between procedures due to insufficient coding, while adjudication resulted in significantly higher data completeness for the majority of the variables.

## 1. Introduction

Danish registries are esteemed for their high quality, with their methodologies and accuracy extensively documented in the existing literature [1,2,3,4,5,6,7,8,9,10,11,12,13,14,15], especially for cardiovascular research [16,17,18,19,20,21,22,23,24,25]. Every resident in Denmark receives a unique personal identification number that enables the linkage of individual data across registries. The Western Denmark Heart Registry (WDHR) has been firmly established as an integral component of the Danish national healthcare and social security system since their inception in 1999 [3,4,8]. The WDHR contains comprehensive information on all cardiac interventions in Western Denmark [1,3,4,26,27]. Currently, only the three major cardiac centers, Aarhus University Hospital, Odense University Hospital, and Aalborg University Hospital, remain in this region. The database is a substantial part in the national monitoring of cardiac intervention quality and provides invaluable, detailed longitudinal data on patients and procedures.

The combination of these robust databases in a population-based healthcare system has paved the way for comprehensive research possibilities, primarily due to the complete follow-up for medical events permitted by linkage with multiple medical and social databases [2,3,8,9,10,11,12,13]. The surgical classification codes used in The Danish National Patient Registry are provided in the Health Care Classification System and is called the Sundhedsvaesenets Klassifikations System (SKS). The SKS is continually developed and maintained by the Danish Health Data Authority.

Although these databases hold invaluable data and have undergone validation, the SKS codes are sometimes insufficient to differentiate between distinct technical methodologies employed within a single type of surgical procedure, such as aortic root replacement (ARR). Notable examples of ARR include aortic valve-sparing root replacements, composite root replacement with mechanical or bioprosthetic valves, root replacement with a Freestyle™ Aortic Root Bioprosthesis (Medtronic, Minneapolis, MN, USA), and root repair using the Florida sleeve method.

Before conducting studies on patients that underwent ARR in Western Denmark, quality and usability of the data were undertaken for two key reasons. First, the WDHR underwent updates in 2003, 2006, and a major update in 2010, each of which carries a potential risk of data inconsistencies. Second, identifying patients who underwent ARR is challenging due to delays in coding updates when new surgical procedures are introduced. As ARR encompasses up to five distinct types of procedures, differentiating between them adds to the complexity.

Our objective was to identify all patients in Western Denmark who underwent ARR and to evaluate the validity of data regarding baseline characteristics and EuroSCORE II variables, along with postoperative outcomes for these patients within the Western Danish Heart Registry, using the electronic medical record as a reference standard.

## 2. Materials and Methods

### 2.1. Study Population

We considered all patients registered in the WDHR from its inception in January 1999 up until April 2022. Inclusion was limited to those with a valid Danish Central Personal Registration (CPR) number, thus excluding non-residents.

Our investigation focused specifically on patients who underwent ARR procedures, as determined by the operation descriptions in the electronic medical records. Cases were classified as ARR if the description indicated root replacement with or without aortic valve replacement and included reimplantation of the coronary arteries, except in the case of Florida sleeve procedures.

### 2.2. Data Sources

The WDHR is a comprehensive, clinical database, which has undergone previous validation, yielding satisfactory levels of quality and completeness [3,8,28]. This registry contains information on all cardiac procedures, including coronary angiography, computed tomography angiography, percutaneous coronary intervention, coronary artery bypass grafting (CABG), heart valve replacement, and transcatheter aortic valve implantation, conducted on patients aged ≥15 years throughout Western Denmark. Surgical data are submitted to the database by individual surgeons at each of the three respective cardiac surgery centers.

Patients’ electronic medical record served as a reference standard in this study. These records offer access to various IT systems containing healthcare staff’s notes, charts, medication, treatment plans, test results, and more. Electronic medical record systems were made mandatory for use in 2004 by the Danish government [29,30]. Efforts were made to digitize older paper-based records, but this process was not consistently comprehensive across all regions or institutions [29,30].

### 2.3. Main Variables

The main variables included all baseline characteristics, EuroSCORE II characteristics, and postoperative outcomes in the WDHR as of April 2022 (see Table 1, Table 2 and Table 3).

### 2.4. Data Collection and Electronic Medical Record Review

The initial data extraction included all procedures classified with SKS codes “KFK” (operations on the mitral valve), “KFC” (operations on the thoracic and thoracoabdominal aorta), “KFM” (operations on the aortic valve), and “KFW” (reoperations following surgery on the heart and major intrathoracic vessels). The electronic medical record of every potential case with SKS codes KFC and KFM, all maintained in electronic format, were thoroughly reviewed. This comprehensive review was performed by a team comprising physicians in cardiac surgery and specifically trained medical students. The medical students had ongoing access to supervision and consultation from specialists in cardiac surgery and cardiology. Following this thorough evaluation, cases were definitively categorized as either “case” or “non-case” (see Figure 1).

**Table 1 diagnostics-15-00611-t001:** Baseline characteristics.

WDHR Registry		Total Patients with Recorded Values		Events Registered		
Characteristics, *N* = 847	Missing Data %	WDHR*N* (%)	Electronic Medical Record *N* (%)	WDHR *N* (%)	Electronic Medical Record *N* (%)	Unrecovered Data %
Angina pectoris	3.7	816 (96.3)	840 (99.2)	77 (9.4)	303 (36.1)	0.8
CCS 1–4 (if angina yes, *n* = 70/320)	3.2	50 (64.9)	282 (93.1)	8/27/9/6	128/68/22/64	6.9
Diabetes mellitus	8.0	779 (92.0)	809 (95.5)	41 (5.3)	40 (4.9)	4.5
Dialysis	8.6	774 (91.4)	840 (99.2)	10 (1.3)	11 (1.3)	0.8
Family history of ischemic heart disease	21.7	663 (78.3)	777 (91.7)	141 (21.3)	183 (23.6)	8.3
Lipid-lowering therapy	10.9	755 (89.1)	809 (95.5)	166 (22.0)	172 (21.3)	4.5
Previous acute myocardial infarction	9.4	767 (90.6)	813 (96.0)	27 (3.5)	32 (3.9)	4.0
Previous percutaneous coronary intervention	9.3	768 (90.7)	813 (96.0)	28 (3.6)	30 (3.7)	4.0
Previous heart surgery	6.1	795 (93.9)	844 (99.6)	98 (12.3)	107 (12.7)	0.4
Previous heart surgery with CABG (if previous yes, *n* = 106/119)	4.7	58 (59.2)	99 (92.5)	8 (13.8)	13 (13.1)	7.5
Smoking status (Never/Previous/Active)	22.1	660 (77.9)	786 (92.8)	300/229/131	324/307/147	7.2
Hypertension therapy	10.3	760 (89.7)	810 (95.6)	380 (50.0)	405 (50.0)	4.4
Arrhythmias	6.0	796 (94.0)	846 (99.9)	108 (13.6)	164 (19.4)	0.1
Atrial fibrillation				79 (9.9)	133 (15.7)	
Block				11 (1.4)	19 (2.2)	
Ventricular fibrillation/tachycardia				7 (0.9)	9 (1.1)	
Other arrhythmias				14 (1.8)	13 (1.5)	
				Mean (SD)	Mean (SD)	
Serum creatinine *, micromol/L	19.8	679 (80.2)	763 (90.1)	96.2 (74.0)	98.7 (85.1)	9.9
Creatinine Clearance *, mL/min	20.9	670 (79.1)	711 (83.9)	97.9 (81.7)	93.8 (57.3)	16.1
Left ventricle ejection fraction, %	14.3	726 (85.7)	804 (94.9)	55.2 (10.8)	55.0 (10.5)	5.1
Height, cm	15.3	717 (84.7)	751 (88.7)	178.3 (9.4)	178.5 (9.4)	11.3
Weight, kg	15.3	717 (84.7)	760 (89.7)	85.3 (16.2)	85.0 (16.3)	10.3
Body mass index, kg/m^2^	15.6	715 (84.4)	748 (88.3)	26.8 (4.4)	26.7 (4.4)	11.7
Body surface area, m^2^	15.6	715 (84.4)	748 (88.3)	2.03 (0.21)	2.03 (0.21)	11.7

Abbreviations: CCS, Canadian Cardiovascular Society grading of angina pectoris; WDHR, Western Danish Heart Registry; CABG, coronary artery bypass grafting. * Blood sample within 30 days of surgery.

**Table 2 diagnostics-15-00611-t002:** EuroSCORE II Characteristics.

WDHR Registry		Total Patients with Recorded Values		Events Registered		
Characteristics, *N* = 847	Missing Data %	WDHR *N* (%)	Electronic Medical Record *N* (%)	WDHR *N* (%)	Electronic Medical Record *N* (%)	Unrecovered Data %
COPD	4.7	807 (95.3)	810 (95.6)	51 (6.3)	52 (6.4)	4.4
PAD	1.4	835 (98.6)	838 (98.9)	66 (7.9)	68 (8.1)	1.1
Poor mobility	3.8	815 (96.2)	819 (96.7)	39 (4.8)	41 (5.0)	3.3
Previous Cardiac Surgery	1.5	834 (98.5)	837 (98.8)	100 (12.0)	101 (12.1)	1.2
Active endocarditis	0.8	840 (99.2)	843 (99.5)	36 (4.3)	44 (5.2)	0.5
Critical preoperative state	3.7	816 (96.3)	818 (96.6)	138 (16.9)	139 (17.0)	3.4
Renal Impairment (Normal/Moderate/Severe/Dialysis)	19.5	682 (80.5)	765 (90.3)	430/186/56/10	490/198/66/11	9.7
CCS angina class 4	1.5	834 (98.5)	838 (98.9)	18 (2.2)	79 (9.4)	1.1
LV function (>50%, 31–50%, 21–30%, ≤20%)	8.6	774 (91.4)	774 (91.4)	630/107/18/11	633/106/19/11	8.6
Recent MI	4.3	811 (95.7)	814 (96.1)	17 (2.1)	17 (2.1)	5.9
Pulmonary hypertension (No/Moderate/Severe)	3.5	817 (96.5)	825 (97.4)	760/37/20	767/37/21	2.6
NYHA classification (I, II, III, IV)	48.9	433 (51.1)	572 (67.5)	111/185/104/33	175/216/148/33	32.5
Surgery on Thoracic Aorta	0.5	843 (99.5)	847 (100)	723 (85.8)	847 (100.0)	0.0
Urgency of operation (Elective/Urgent/Emergency/Salvage)	5.7	799 (94.3)	805 (95.0)	527/36/198/38	530/37/195/43	5.0
Weight of operation—3 procedures	0.5	843 (99.5)	847 (100)	148 (17.6)	842 (99.4)	0
EuroSCORE II, median (IQR)				6.21 (1.42–6.30)	10.7 (3.32–13.3)	

Abbreviations: COPD, chronic obstructive pulmonary disease; PAD, peripheral artery disease; CCS, Canadian Cardiovascular Society grading of angina pectoris; LV, left ventricular; MI, myocardial infarction; NYHA, New York Heart Association.

**Table 3 diagnostics-15-00611-t003:** Operative Variables.

WDHR Registry		Total Patients with Recorded Values		Events Registered			
Characteristics, *N* = 847	Missing Data %	WDHR *N* (%)	Electronic Medical Record *N* (%)	WDHR *N* (%)	Electronic Medical Record *N* (%)	PPV, % (95% CI)	Unrecovered Data %
Reoperation for bleeding	4.1	812 (95.9)	844 (99.6)	113 (13.9)	132 (15.6)	96.5 (91.2–99.0)	0.4
Reoperation for ischemia	8.1	778 (91.9)	844 (99.6)	9 (1.2)	12 (1.4)	100 (66.4–100)	0.4
Intra-aortic balloon pump	4.8	806 (95.2)	845 (99.8)	16 (2.0)	19 (2.2)	93.8 (69.8–99.8)	0.2
Assist device	4.5	809 (95.5)	845 (99.8)	24 (3.0)	26 (3.1)	91.3 (72.0–98.9)	0.6
				Mean (SD)	Mean (SD)		
Extra corporeal circulation, min	9.0	771 (91.0)	775 (91.5)	215.4 (102.8)	215.3 (103.1)		8.5
Aortic clamp time, min	9.1	770 (90.9)	775 (91.5)	136.1 (52.8)	135.6 (53.5)		8.5
Maximal CK-MB, µg/L	35.1	550 (64.9)	617 (72.8)	63.6 (108.6)	63.0 (105.5)		27.2

Abbreviations: CK-MB, creatine kinase-myocardial band.

### 2.5. Validation

In our validation process, each data point extracted from the database was carefully adjudicated by cross-referencing them against its corresponding entry within the electronic medical record. The electronic medical record served as the definitive reference in this process, providing a reliable reference for data verification.

### 2.6. Statistical Analysis

Continuous variables were reported as mean (standard deviation, SD) for normally distributed data, or median (range: minimum–maximum) for non-normally distributed data, with distribution shape assessed using histograms. Numbers and respective percentages were calculated for categorical variables. Group comparisons were performed accordingly with Student’s *t*-test (alternatively: Wilcoxon rank sum test) and Chi-squared test (alternatively: Fisher’s exact test). EuroSCORE II was calculated for each patient by integrating all requisite parameters into the established formula, and 95% confidence intervals (CI) were reported as appropriate [31]. Patients with missing data did not have a EuroSCORE II calculation. Weight of operation for ARR was three procedures, including surgery on the aortic valve, the thoracic aorta, and the coronary arteries. Missing data were treated as representing either absent or normal values (e.g., if the diagnosis diabetes mellitus was never noted in the electronic medical record, and no anti-diabetic medicine was prescribed, the case was adjudicated as non-diabetic). We calculated the positive predictive value (PPV) for postoperative outcomes. PPV was calculated as the ratio of confirmed events according to the electronic medical record to the total number of registered events in the WDHR. Point estimates are supplemented by respective 95% CI as appropriate. A *p*-value of less than 0.05 is considered statistically significant. Statistical analysis was performed with STATA/IC 18 (StataCorp, College Station, TX 77845, USA).

## 3. Results

### 3.1. Aortic Root Replacement Cases in Western Denmark

A total of 22,247 cases in the WDHR registered with aortic valve surgery, mitral valve surgery, or other surgery coded under SKS “KFK”, “KFC”, “KFM”, or “KFW” were identified between January 1999 and April 2022. The flowchart (Figure 1) provides a detailed overview of the steps leading to the final study population, including the identification, exclusion criteria, and validation process. After reviewing 10,362 patients, we identified 847 (8.2%) confirmed cases of ARR. Of the 847 cases, there were 568 cases of composite root replacement with mechanical valves, 137 cases of composite root replacement with bioprosthetic valves, and 52 cases with Freestyle™ Aortic Root Bioprosthesis (Medtronic, Minneapolis, MN, USA). Lastly, 90 cases of aortic valve-sparing root replacement were recorded, of which 80 were David procedures, five Yacoub procedures, and five Florida sleeve procedures. The primary indications for surgery with ARR in the study period were isolated thoracic aortic aneurism (TAA) (139 (16.4%)), isolated aortic regurgitation (AR) (17 (2.0%)), TAA with concomitant AR (380 (44.9%)), aortic dissection (228 (26.9%)), and infectious endocarditis (44 (5.2%)). The remaining 39 patients were operated on due to aortic stenosis, thrombus formation, connective tissue disease, and subvalvular stenosis.

### 3.2. Baseline Characteristics

The baseline characteristics in the WDHR had missing data for 3.2% to 22.1%, with an average of 12.0% missing data per variable in our study population (Table 1). Between 23% and 98% of missing data for each baseline variable in the WDHR were recovered by adjudication of data with the electronic medical record. After adjudication, missing data for baseline characteristics ranged from 0.1% to 11.7% (mean 6.5%), and 18 out of 24 variables had a significantly higher completeness compared to the WDHR dataset.

### 3.3. EuroSCORE II Characteristics

The EuroSCORE II variables had missing data ranging from 0.8% to 48.9%, with an average of 7.3% missing per variable (Table 2). The variables with the highest rates of missing data were renal impairment (19.5%), left ventricle function (8.6%), NYHA classification (48.9%), and urgency of operation (5.7%). After adjudication, missing data for EuroSCORE II variables ranged from 0% to 32.5% (mean 5.3%), and all variables showed significantly higher completeness compared to the WDHR dataset. Furthermore, missing data for each EuroSCORE II variable in the WDHR were recovered, ranging from 6% to complete recovery (100%) (Table 2). Missing data for EuroSCORE II parameters among patients registered from 2012 until the end of the study were generally lower, with a range from 0.6% to 14.3% (mean 2.9%), while completeness of data after adjudication was significantly higher in 4 out of 15 variables. The procedure weight classified as 3 was reported for 148 patients (17.6%) at the index event and 842 patients (99.4%) after adjudication. The median EuroSCORE II according to the WDHR was 6.21% (95% CI 1.42–6.30), which increased significantly to 10.7% (95% CI 3.32–13.3) after adjudication.

### 3.4. Operative Variables and Postoperative Outcomes

Lastly, categorical operative variables and postoperative outcome variables had missing data ranging from 4.1% to 8.1%, with an average of 5.5% missing data per variable (Table 3 and Table 4). Continuous postoperative outcome variables had missing data between 9.0% and 35.1%. After adjudication, missing data for categorical postoperative outcomes ranged from 0% to 0.8% (mean 0.5%) and for continuous variables from 8.5% to 27.2%. All variables showed significantly higher completeness compared to the original dataset. Furthermore, missing data for each postoperative outcome in the WDHR were recovered, ranging from 7% to complete recovery (100%). The PPV was calculated for each postoperative outcome, ranging between 59 and 100%. Postoperative atrial fibrillation (POAF) was identified in 411 (48.7%) patients. PPV for postoperative atrial fibrillation (POAF) was 89.2 (95% CI 84.7–92.8); PPV values for all arrhythmia, central nervous damage, dialysis, reoperation for bleeding, and reoperation for ischemia exceeded 95%.

## 4. Discussion

In this comprehensive review of patients undergoing ARR as registered in the WDHR, we identified three key findings.

First, only 8.2% of the reviewed patients underwent ARR. This was primarily due to the broad coding criteria used; however, some cases also reflected the development of surgical procedures and the evolution of coding systems over time. Validation of these complex surgical procedures is evidently necessary to ensure that all procedures are detected and subdivided correctly, especially for aortic valve-sparing root replacement.

Second, the WDHR demonstrates overall value for clinical epidemiological research; however, some baseline characteristics had missing data ranging from 19.8% to 48.9%. Validation of these parameters against the electronic medical record recovered 40–50% of missing data, while 18 out of 24 baseline characteristics along with all EuroSCORE II characteristics and postoperative outcomes were significantly more complete after adjudication with the electronic medical record. Significant improvements in data completeness were also present for EuroSCORE II parameters among patients registered from 2012 and beyond. The discrepancy in missing data between parameters may be attributed to the continuous improvement of the WDHR, in which some of the parameters have become mandatory to report over time. Considering this, missing data might indicate a missing time bias, which researchers should recognize and manage carefully in retrospective studies. Thus, our study demonstrates that adjudication of clinically important parameters with the electronic medical record might be necessary and worth doing to obtain higher data completeness in general, with the majority of variables being significantly more complete after validation. EuroSCORE II was also significantly different between the WDHR and the electronic medical record, which may be due to three reasons: the weight of operation (i.e., number of performed procedures), surgery on the thoracic aorta, and significantly higher data completeness among all EuroSCORE II parameters after adjudication with the electronic medical record. In 699 of the cases (82.5%), ARR procedures were reported as a single or double procedure, respectively. All procedures, except for the five cases with the Florida sleeve, were reported as triple procedures in the electronic medical record (i.e., surgery on the aortic valve, the thoracic aorta, and reimplantation of the coronary arteries). The Florida sleeve only included two procedures because of no surgery on the coronary arteries. Thus, the EuroSCORE II was significantly higher after adjudication, primarily due to the misclassification of procedural weight, and including surgery involving the thoracic aorta. This reassessment likely provides a more accurate representation of the complexity of ARR procedures, as categorizing them as single or double procedures underestimates both their intricacy and the associated risks. However, it is important to note that the number of root replacement cases included in the development of EuroSCORE II was limited [31].

Third, the validity was relatively high for almost all postoperative outcomes, especially central nervous damage, arrythmias, reoperation for bleeding, reoperation for ischemia, dialysis, and intra-aortic balloon pump with PPVs ranging between 93.8 and 100%. The PPV for POAF in our study was 89.2%. This is higher than a previous evaluation of POAF validity in a large cohort from the WDHR, which reported a PPV of 82.5% (95% CI: 78.8–85.7) [26]. However, it is important to note that our validation encompassed all patients undergoing ARR since 1999, whereas the previous study focused on a population undergoing on- or off-pump CABG, mitral or aortic valve surgery, or combined procedures between 1 January 2011 and 31 December 2013. They included a shorter study period due to an update in the WDHR, in which registration of the individual’s arrythmia became mandatory from January 2011. After validation, Munkholm et al. [26] found that 490 patients developed POAF, equivalent to 36.4% of the population, while 205 patients had non-corresponding registrations of atrial fibrillations. In our study, we found that 48.1% patients developed POAF, while validation led to the discovery of 161 additional cases (39.2%). These findings further emphasize the need for validation of clinical parameters, which have changed over time within the WDHR, as exemplified with POAF. In our study, this is specifically due to the risk of missing time bias, which also may explain the significantly higher data completeness for postoperative outcomes. To improve the quality of the WDHR, possible measures could include enhancing data validation protocols, ensuring more regular registry updates, and considering the integration of adjudication processes into routine data management.

Considering these key findings, our validation of ARR procedures within the WDHR have provided a robust multicenter database with remarkably high data completeness due to adjudication of all registered data. This achievement offers extensive scientific applications, enabling the investigation of key research questions and delivering reliable conclusions, while also fostering possible national and international collaborations on ARR procedures.

## 5. Perspectives

This adjudicated data serves as the foundation for future cohort studies aimed at improving the quality of research on patients undergoing ARR. Our goal is to include all cases of ARR from the fourth cardiac center in Denmark, Rigshospitalet, expanding the dataset to provide nationwide coverage for these increasingly frequent surgical procedures, which require comprehensive analysis.

## 6. Limitations

Our study has several strengths and limitations. It is the first study to evaluate several parameters within the WDHR for patients who have undergone AAR, including baseline and EuroSCORE II variables along with postoperative outcomes. Thus, the results stem from a specifically selected cohort of patients within the WDHR, which limits our findings to this specific patient group. We analyzed data from January 1999 to April 2022, covering all data on this topic since the inception of the WDHR, and compared their validity to the electronic medical record.

The electronic medical record was reviewed and chosen as the reference standard, which may represent a possible limitation since our findings depend on the quality and accuracy of the electronically available information. We had access to all electronic patient data, including lab results and diagnosis codes. The electronic medical record was implemented in the Danish healthcare system during the first decade with efforts to digitalize paper-based records. Though there is a risk of missing data bias in the early part of our study period, substantial progress in data collection was still made during this time. Furthermore, we did not have access to data from general practitioners, but this may not have had a significant impact on our findings due to comprehensive perioperative reporting. All postoperative endpoints are characterized as hard endpoints within 30 days, meaning that postoperative events would likely have led to hospitalization and thus registration of the event in the electronic medical record. Lastly, EuroSCORE II was released in 2012 [31], making its applicability to patients operated on prior to this date uncertain; however, its development was based on patients treated before 2012.

This presents an important limitation, contributing to missing time bias, as surgeons did not register EuroSCORE II variables when only EuroSCORE I was required. Additionally, we only calculated the score for patients without missing data, which introduces potential bias in the mean calculation. However, these patients are underrepresented in the EuroSCORE II cohort, raising concerns about the reliability of using EuroSCORE II as a risk score for this population.

## 7. Conclusions

The WDHR demonstrates considerable value for clinical epidemiological research. However, patient identification required extensive review of medical records due to insufficient coding with SKS codes. The use of electronic medical records for adjudication also led to a significantly higher data completeness for almost all variables in the WDHR, but missing values in EuroSCORE II variables post-adjudication were generally lower after 2012. Thus, our efforts to validate and adjudicate all data from the WDHR have provided a multicenter database on ARR with significantly high data completeness, positively affecting data reliability and, consequently, the future robustness of research conclusions derived from these data.

## Figures and Tables

**Figure 1 diagnostics-15-00611-f001:**
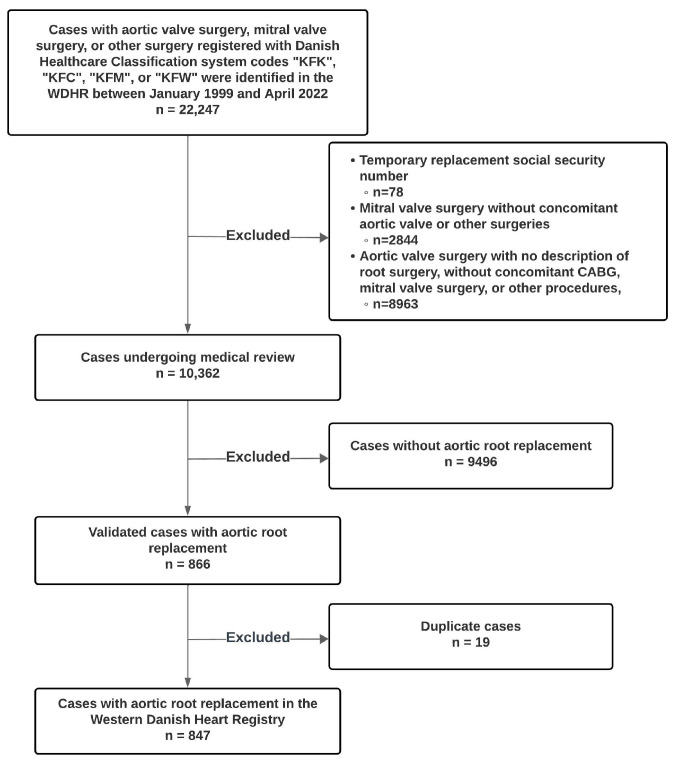
Flowchart of Patient Selection and Inclusion Criteria. Abbreviations: WDHR, Western Danish Heart Registry; CABG, Coronary Artery Bypass Grafting.

**Table 4 diagnostics-15-00611-t004:** Postoperative Outcomes.

**WDHR Registry**		**Total Patients with Recorded Values**		**Events Registered**			
**Characteristics, *N* = 847**	**Missing Data** **%**	**WDHR** ***N* (%)**	**Electronic Medical Record** ***N* (%)**	**WDHR** ***N* (%)**	**Electronic Medical Record** ***N* (%)**	**PPV, % ** **(95% CI)**	**Unrecovered Data %**
Central nervous damage	5.8	798 (94.2)	844 (99.6)	58 (7.3)	66 (7.8)	98.3 (90.7–99.9)	0.4
Sternal infection	5.3	802 (94.7)	845 (99.8)	10 (1.2)	10 (1.2)	70.0 (34.8–93.3)	0.7
Arrhythmia	4.4	810 (95.6)	844 (99.6)	327 (40.4)	506 (60.6)	99.1 (97.3–99.8)	0.4
Atrial fibrillation (if arrhythmia yes)	4.6	810 (100)	844 (100)	250 (30.9)	411 (48.7)	89.2 (84.7–92.8)	0.0
Atrioventricular block (if arrhythmia yes)	4.6	810 (100)	838 (99.3)	62 (7.7)	109 (13.7)	87.3 (76.5–94.4)	0.7
Reoperation for bleeding	4.1	812 (95.9)	844 (99.6)	113 (13.9)	132 (15.6)	96.5 (91.2–99.0)	0.4
Reoperation for ischemia *	8.1	778 (91.9)	844 (99.6)	9 (1.2)	12 (1.4)	100 (66.4–100)	0.4
Dialysis	7.1	787 (92.9)	840 (99.2)	79 (10.0)	88 (10.5)	97.5 (91.2–99.7)	0.8

* Reoperation for ischemia refers to cases where elevated CK-MB and/or troponin levels, and/or coronary angiography findings indicating significant coronary obstruction, necessitating surgical intervention.

## Data Availability

Data are not publicly available but will be made available upon reasonable request. Data will be shared whose proposed use has been approved, primarily for replication of our results. A signed data access agreement, compliant with regional legislation and data authority requirements, is required prior to data release. To request access, contact Lytfi.Krasniqi@rsyd.dk.

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
