# Peer review of "Aortic Root Replacement Procedures: A Validation Study of the Western Denmark Heart Registry from 1999 to 2022"

_diagnostics, 2025, doi:10.3390/diagnostics15050611_

Round 1
Reviewer 1 Report
Comments and Suggestions for Authors
I read with great interested the paper "Aortic root replacement procedures: A Validation Study of the Western Denmark Heart Registry from 1999-2022" by Dr. Ravn and colleagues.
The manuscript is well written and can be clearly read throughout the whole manuscript. Limitations are correctly stated.
This is a particular paper focusing on the quality of a national database from Denmark. The Authors have highlighted the problems related to such registries, particularly occurring after mandatory data updates. After adjudication, more reliable data was created.
The manuscript could benefit from the following comments:
Methods
- The meaning of the KFC, KFM, KFK, and KFW codes should be clarified in the Method section.
Results
- There was quite a change in EuroSCORE II from the WDHR to after adjudication. However this was elegantly discussed in the Discussion section. This is a proof of how updates of registries can influence the data.
- Was the data of the WDHR updated after adjudication to improve data quality?
Discussion
- Considering all the comments the Authors have provided and all the problems have been raised regarding this database, I would recommend adding a paragraph with possible solutions and their feasibility to improve the quality of the WDHR. This can be taken as strong example for other countries.
Author Response
Comment 1:
The manuscript is well written and can be clearly read throughout the whole manuscript. Limitations are correctly stated. This is a particular paper focusing on the quality of a national database from Denmark. The Authors have highlighted the problems related to such registries, particularly occurring after mandatory data updates. After adjudication, more reliable data was created. The manuscript could benefit from the following comments:
Response 1:
We sincerely appreciate the reviewer’s positive feedback on the clarity and quality of our manuscript. We are pleased that the limitations have been clearly conveyed. Additionally, we appreciate the recognition of our adjudication process in enhancing data reliability. Please find the detailed responses to the specific comments raised and have revised the manuscript accordingly to further improve its clarity of our manuscript.
********************************
Comment 2:
Methods: The meaning of the KFC, KFM, KFK, and KFW codes should be clarified in the Method section.
Response 2:
Thank you for your valuable comment. We have, accordingly, clarified the meaning of the KFC, KFM, KFK, and KFW codes in the Methods section. Please see lines 106-109 in the revised manuscript:
Data Collection and Electronic Medical record Review
The initial data extraction included all procedures classified with SKS codes “KFK” (operations on the mitral valve), “KFC” (operations on the thoracic and thoracoabdominal aorta), “KFM” (operations on the aortic valve), and “KFW” (reoperations following surgery on the heart and major intrathoracic vessels). The electronic medical record of every potential case with SKS codes KFC and KFM, all maintained in electronic format, were thoroughly reviewed. This comprehensive review was performed by a team comprised of physicians in cardiac surgery and specifically trained medical students. The medical students had ongoing access to supervision and consultation from specialists in cardiac surgery and cardiology. Following this thorough evaluation, cases were definitively categorized as either 'case' or 'non-case' (see Figure 1).
********************************
Comment 3:
Results: There was quite a change in EuroSCORE II from the WDHR to after adjudication. However this was elegantly discussed in the Discussion section. This is a proof of how updates of registries can influence the data. Was the data of the WDHR updated after adjudication to improve data quality?
Response 3:
Thank you for this important comment. The WDHR data was not updated after adjudication as we did not have permission to make such modifications to the WDHR. This study was conducted independently of the WDHR board although our co-author LPR is a current board member and PEM is a former chair of the WDHR.
********************************
Comment 4:
Discussion: Considering all the comments the Authors have provided and all the problems have been raised regarding this database, I would recommend adding a paragraph with possible solutions and their feasibility to improve the quality of the WDHR. This can be taken as strong example for other countries.
Response 4:
Thank you for your insightful suggestion. We have added a paragraph to the Discussion section outlining potential solutions to enhance the quality and feasibility of WDHR.
Please see lines 254-257 in the revised manuscript:
To improve the quality of the WDHR, possible measures could include enhancing data validation protocols, ensuring more regular registry updates, and considering the integration of adjudication processes into routine data management.
Reviewer 2 Report
Comments and Suggestions for Authors
I have carefully read the article titled "Aortic root replacement procedures: A Validation Study of the 2 Western Denmark Heart Registry from 1999-2022" sent to me for evaluation. My comments, criticisms and suggestions are listed below:
First of all, I would like to congratulate the authors for this clinical epidemiological study, which has been worked on very hard. The authors' efforts to verify and decide on the data from WDHR have resulted in the creation of a multicenter database with increased data integrity. Increased data reliability will help studies to be conducted based on this data to produce more accurate and useful results. Another important point here is that the main variables are selected correctly.
Some changes should be made, especially in Table 3: It would be good to add important variables such as hospital stay, intensive care stay, mechanical ventilation time, amount of blood and blood products used, renal failure not requiring dialysis, prolonged ventilation or pulmonary failure, multiorgan failure, low cardiac output, need for inotropic support to this table, which includes postoperative variables. What is meant by “Reoperation for ischemia”? It is not understood. The title of Table 3 is “Postoperative Outcomes”, but some operative variables are included in this table (extracorporeal circulation time, aortic clamp time). It would be appropriate for these variables to be included in a separate table under the title “Operative Variables”. It would be appropriate to add important variables such as thoracotomy incision type, arterial cannulation location, duration of circulatory arrest if performed, additional valve procedures or additional coronary artery bypass procedures, and valve prosthesis type used to the operative variables table. Thus, it will be possible for studies based on this database to reach clearer scientific results.
Best Regards
Author Response
Comment 1:
First of all, I would like to congratulate the authors for this clinical epidemiological study, which has been worked on very hard. The authors' efforts to verify and decide on the data from WDHR have resulted in the creation of a multicenter database with increased data integrity. Increased data reliability will help studies to be conducted based on this data to produce more accurate and useful results. Another important point here is that the main variables are selected correctly.
Response 1:
Thank you for recognizing and acknowledging our efforts to enhance data integrity in the WDHR. We appreciate the time you took to review and comment on our manuscript. Your feedback is greatly valued, and we hope we have adequately addressed all your comments.
********************************
Comment 2:
Some changes should be made, especially in Table 3: It would be good to add important variables such as hospital stay, intensive care stay, mechanical ventilation time, amount of blood and blood products used, renal failure not requiring dialysis, prolonged ventilation or pulmonary failure, multiorgan failure, low cardiac output, need for inotropic support to this table, which includes postoperative variables.
Response 2:
Thank you for this valuable suggestion. Unfortunately, we do not have access to all these variables. While some, such as hospital stay and intensive care stay, may be registered in the WDHR, they were not part of the adjudication process. We agree that including these variables would further enhance the quality of our database and provide a more comprehensive overview of postoperative outcomes.
********************************
Comment 3:
What is meant by “Reoperation for ischemia”? It is not understood
Response 3:
Thank you for your comment. Reoperation for ischemia refers to cases where elevated CK-MB and/or troponin levels, along with coronary angiography findings indicating significant coronary obstruction, necessitate surgical intervention. No specific cutoff values are used in daily clinical practice, instead, each case is thoroughly evaluated by cardiac surgeons, anesthesiologists, and cardiologists, and individualized decisions are made based on the clinical context.
We have clarified this in the revised manuscript. Please see table 4 for the updated definition:
Reoperation for ischemia refers to cases where elevated CK-MB and/or troponin levels, and/or coronary angiography findings indicating significant coronary obstruction, necessitate surgical intervention.
********************************
Comment 4:
The title of Table 3 is “Postoperative Outcomes”, but some operative variables are included in this table (extracorporeal circulation time, aortic clamp time). It would be appropriate for these variables to be included in a separate table under the title “Operative Variables”.
Response 4:
Thank you for your suggestion. We have revised the manuscript accordingly and created a new table titled "Operative Variables" to separate these from the "Postoperative Outcomes" table.
********************************
Comment 5:
It would be appropriate to add important variables such as thoracotomy incision type, arterial cannulation location, duration of circulatory arrest if performed, additional valve procedures or additional coronary artery bypass procedures, and valve prosthesis type used to the operative variables table. Thus, it will be possible for studies based on this database to reach clearer scientific results.
Response 5:
Thank you for your suggestion. Unfortunately, thoracotomy incision type is not recorded in the WDHR. While the remaining variables would have been valuable additions, we did unfortunately not adjudicate them. However, based on our experience, these variables are generally well-documented in the registry, which is why we primarily focused on identifying patients undergoing root replacement and adjudicating patient characteristics and outcomes. Future studies will indeed incorporate these variables from the WDHR.